# Hallmarks of Hunds coupling in the Mott insulator $Ca_2RuO_4$

D. Sutter[1,*], C.G. Fatuzzo[2,*], S. Moser[3], M. Kim[4,5], R. Fittipaldi[6,7], A. Vecchione[6,7], V. Granata[6,7], Y. Sassa[8], F. Cossalter[1], G. Gatti[2], M. Grioni[2], H.M. Rønnow[2], N.C. Plumb[9], C.E. Matt[9], M. Shi[9], M. Hoesch[10], T.K. Kim[10], T.-R. Chang[11,12], H.-T. Jeng[11,13], C. Jozwiak[3], A. Bostwick[3], E. Rotenberg[3], A. Georges[4,5,14], T. Neupert[1] & J. Chang[1]

A paradigmatic case of multi-band Mott physics including spin-orbit and Hund's coupling is realized in $Ca_2RuO_4$. Progress in understanding the nature of this Mott insulating phase has been impeded by the lack of knowledge about the low-energy electronic structure. Here we provide—using angle-resolved photoemission electron spectroscopy—the band structure of the paramagnetic insulating phase of $Ca_2RuO_4$ and show how it features several distinct energy scales. Comparison to a simple analysis of atomic multiplets provides a quantitative estimate of the Hund's coupling $J = 0.4\,eV$. Furthermore, the experimental spectra are in good agreement with electronic structure calculations performed with Dynamical Mean-Field Theory. The crystal field stabilization of the $d_{xy}$ orbital due to $c$-axis contraction is shown to be essential to explain the insulating phase. These results underscore the importance of multi-band physics, Coulomb interaction and Hund's coupling that together generate the Mott insulating state of $Ca_2RuO_4$.

[1] Physik-Institut, Universität Zürich, Winterthurerstrasse 190, Zürich CH-8057, Switzerland. [2] Institute of Physics, École Polytechnique Fedérale de Lausanne (EPFL), Lausanne CH-1015, Switzerland. [3] Advanced Light Source (ALS), Berkeley, California 94720, USA. [4] College de France, Paris Cedex 05 75231, France. [5] Centre de Physique Théorique, Ecole Polytechnique, CNRS, Univ Paris-Saclay, Palaiseau 91128, France. [6] CNR-SPIN, Fisciano, Salerno I-84084, Italy. [7] Dipartimento di Fisica 'E.R. Caianiello', Università di Salerno, Fisciano, Salerno I-84084, Italy. [8] Department of Physics and Astronomy, Uppsala University, Uppsala S-75121, Sweden. [9] Swiss Light Source, Paul Scherrer Institut, Villigen PSI CH-5232, Switzerland. [10] Diamond Light Source, Harwell Campus, Didcot OX11 0DE, UK. [11] Department of Physics, National Tsing Hua University, Hsinchu 30013, Taiwan. [12] Department of Physics, National Cheng Kung University, Tainan 701, Taiwan. [13] Institute of Physics, Academia Sinica, Taipei 11529, Taiwan. [14] Department of Quantum Matter Physics, University of Geneva, Geneva 4 1211, Switzerland. * These authors contributed equally to this work. Correspondence and requests for materials should be addressed to D.S. (email: dsutter@physik.uzh.ch) or to J.C. (email: johan.chang@physik.uzh.ch).

Electronic instabilities driving superconductivity, density wave orders and Mott metal–insulator transitions produce a characteristic energy scale below an onset temperature[1–3]. Typically, this energy scale manifests itself as a gap in the electronic band structure around the Fermi level. Correlated electron systems have a tendency for avalanches, where one instability triggers or facilitates another[4]. The challenge is then to disentangle the driving and secondary phenomena. In many Mott insulating systems, such as $La_2CuO_4$ and $Ca_2RuO_4$, long-range magnetic order appears as a secondary effect. In such cases, the energy scale associated with the Mott transition is much larger than that of magnetism. The Mott physics of the half-filled single-band $3d$ electron system $La_2CuO_4$ emerges due to a high ratio of Coulomb interaction to band width. This simple scenario does not apply to $Ca_2RuO_4$. There the orbital and spin degrees of freedom of the 2/3-filled (with four electrons) $t_{2g}$-manifold implies that Hund's coupling enters as an important energy scale[5]. Moreover, recent studies of the antiferromagnetic ground state of $Ca_2RuO_4$ suggest that spin–orbit interaction also plays a significant role in shaping the magnetic moments[6–8], as well as the splitting of the $t_{2g}$ states[9].

Compared to $Sr_2RuO_4$ (refs 10,11), which may realize a chiral $p$-wave superconducting state, relatively little is known about the electronic band structure of $Ca_2RuO_4$ (ref. 12). Angle integrated photoemission spectroscopy has revealed the existence of Ru states with binding energy 1.6 eV (ref. 13)—an energy scale much larger than the Mott gap $\sim 0.4$ eV estimated from transport experiments[14]. Moreover, angle-resolved photoemission spectroscopy (ARPES) experiments on $Ca_{1.8}Sr_{0.2}RuO_4$—the critical composition for the metal–insulator transition—have led to contradicting interpretations[15,16] favouring or disfavouring the so-called orbital-selective scenario where a Mott gap opens only on a subset of bands[17,18]. Extending this scenario to $Ca_2RuO_4$ would imply orbital-dependent Mott gaps[18]. The electronic structure should thus display two Mott energy scales (one of $d_{xy}$ and another for the $d_{xz}$, $d_{yz}$ states). A different explanation for the Mott state of $Ca_2RuO_4$ is that the $c$-axis compression of the S-Pbca insulating phase induces a crystal field stabilization of the $d_{xy}$ orbital, leading to half-filled $d_{xz}$, $d_{yz}$ bands and completely filled $d_{xy}$ states[19,20]. In this case, only one Mott gap on the $d_{xz}$, $d_{yz}$ bands will be present with band insulating $d_{xy}$ states. The problem has defied a solution due to a lack of experimental knowledge about the low-energy electronic structure.

Here we present an ARPES study of the electronic structure in the paramagnetic insulating state (at 150 K) of $Ca_2RuO_4$. Three different bands—labelled $\mathcal{A}$, $\mathcal{B}$ and $\mathcal{C}$ band—are identified and their orbital character is discussed through comparison to first-principle Density Functional Theory (DFT) band structure calculations. The observed band structure is incompatible with a single insulating energy scale acting uniformly on all orbitals. A phenomenological Green's function incorporating an enhanced crystal field and a spectral gap in the self-energy is used to describe the observed band structure on a qualitative level. Further insight is gained from Dynamical Mean-Field Theory (DMFT) calculations including Hund's coupling and Coulomb interaction. The Hund's coupling splits the $d_{xy}$ band allowing quantitative estimate of this parameter. The Coulomb interaction is mainly responsible for the insulating behaviour of the $d_{xz}$, $d_{yz}$ bands. The experimental results, together with our theoretical analysis, clarify the origin of the Mott phase in the multi-orbital system $Ca_2RuO_4$. Furthermore, they provide a natural explanation as to why previous experiments have identified different values for the energy gap.

## Results

**Crystal and electronic structure.** $Ca_2RuO_4$ is a layered perovskite, where the Mott transition coincides with a structural transition at $T_s \sim 350$ K, below which the $c$-axis lattice constant is reduced. We study the paramagnetic insulating state ($T = 150$ K) of $Ca_2RuO_4$ with orthorhombic S-Pbca crystal structure ($a = 5.39$ Å, $b = 5.59$ Å and $c = 11.77$ Å). It is worth noting that due to this nonsymmorphic crystal structure, $Ca_2RuO_4$ could not form a Mott insulating ground state at other fillings than 1/3 and 2/3 (ref. 21). In Fig. 1, the experimentally measured electronic

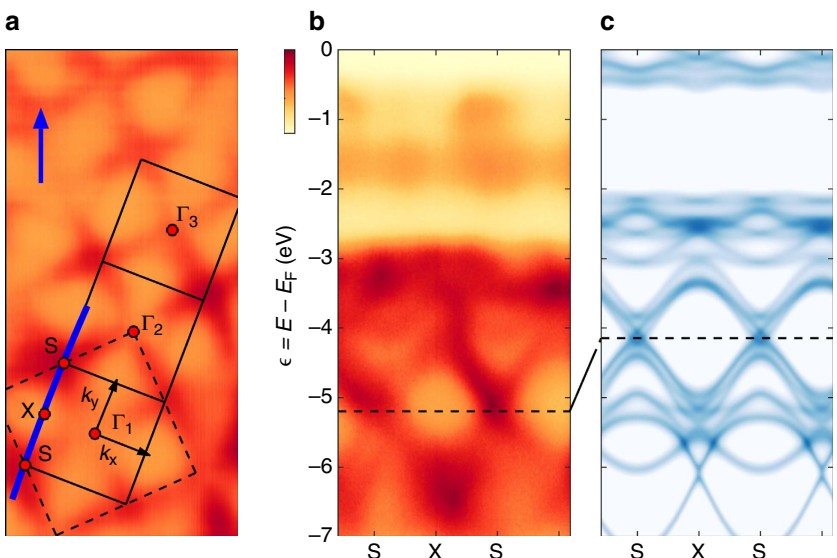

**Figure 1 | Oxygen band structure of $Ca_2RuO_4$.** ARPES recorded with right-handed circularly polarized (C$^+$) 65 eV photons in the paramagnetic (150 K) insulating state of $Ca_2RuO_4$, compared to DFT band structure calculations. Incident direction of the light is indicated by the blue arrow. Dark colours correspond to high intensities. (**a**) Constant energy map displaying the photoemission spectral weight at binding energy $\epsilon = E - E_F = -5.2$ eV. Solid and dashed lines mark the in-plane projected orthorhombic and tetragonal zone boundaries, respectively. $\Gamma_i$ with $i = 1, 2, 3$ label orthorhombic zone centres. S and X label the zone corners and boundaries, respectively. (**b**) Spectra recorded along the zone boundary (blue line in **a**) Oxygen-dominated bands are found between $\epsilon = -7$ and $-3$ eV, whereas the ruthenium bands are located above $-2.5$ eV. (**c**) First-principle DFT band structure calculation. Within an arbitrary shift, indicated by the dashed line, qualitative agreement with the experiment is found for the oxygen bands.

structure is compared to a first-principle DFT calculation of the bare non-interacting bands. We observe two sets of states: near the Fermi level the electronic structure is comprised of Ru-dominated bands, while oxygen bands are present only for $\epsilon = E - E_F < -2.5$ eV. Up to an overall energy shift, good agreement between the calculated DFT and observed $Ca_2RuO_4$ oxygen band structure is found.

**Non-dispersing ruthenium bands.** The structure of the ruthenium bands near the Fermi level is the main topic of this paper, as these are the states influenced by Mott physics. A compilation of ARPES spectra, recorded along high-symmetry directions, is presented in Figs 2 and 3a. In consistency with previous angle-integrated photoemission experiments[13], a broad and flat band is found around the binding energy $\epsilon = -1.7$ eV. However, we also observe spectral weight closer to the Fermi level ($\epsilon \sim -0.8 \pm 0.2$ eV), especially near the zone boundaries (see Fig. 2a,d). These two flat ruthenium bands (labelled $\mathcal{A}$ and $\mathcal{B}$) are revealed as a double peak structure in the energy distribution curves—Fig. 2c,f. Between the $\mathcal{A}$ band and the Fermi level, the spectral weight is suppressed. In fact, complete suppression of spectral weight is found for $-0.2$ eV $< \epsilon < 0$ eV (see Fig. 2c). This energy scale is in reasonable agreement with the activation energy $\sim 0.4$ eV extracted from resistivity experiments[14].

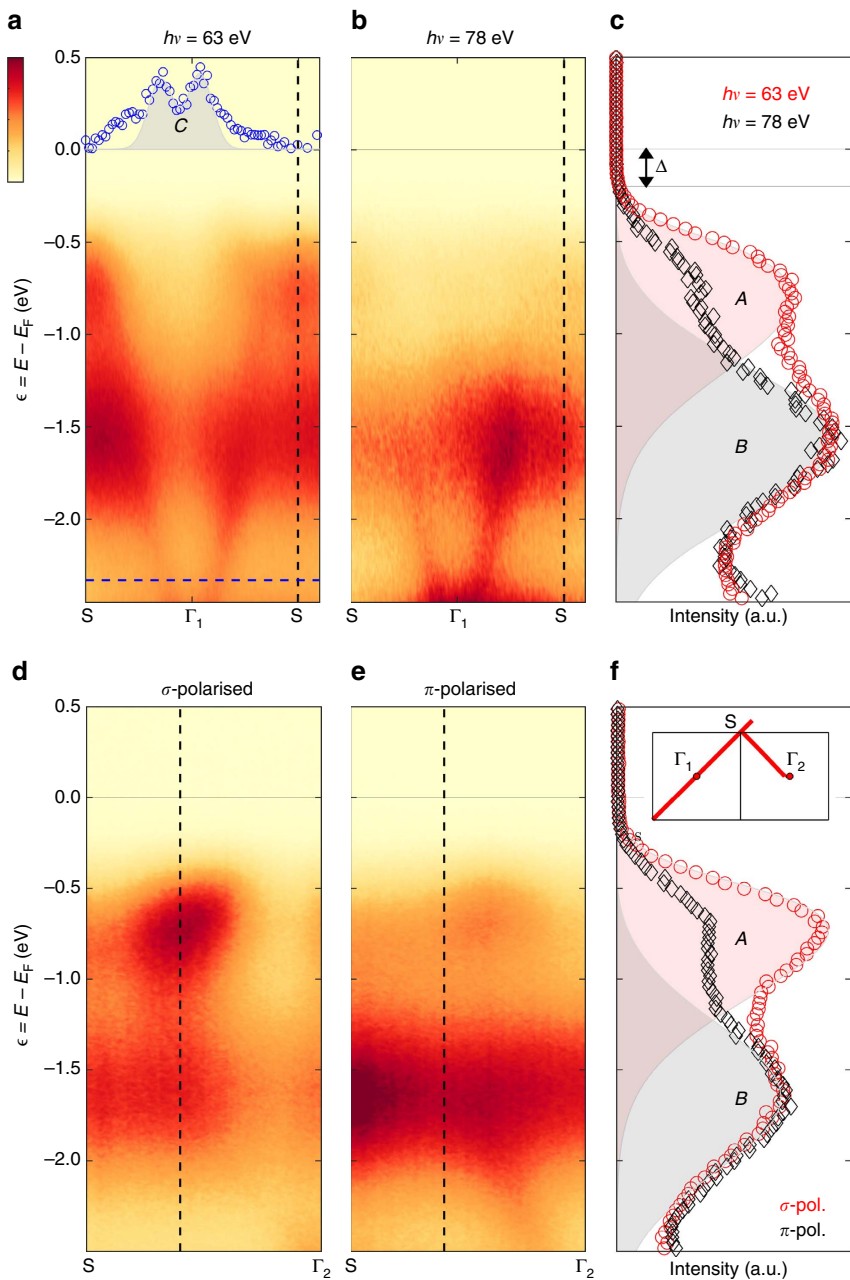

**Figure 2 | Ruthenium band structure.** (**a,b**) Photoemission spectra recorded along the high-symmetry direction $\Gamma_1 - S$ for incident circularly polarized light with photon energies $h\nu$ as indicated. Dark colours correspond to high intensities. Blue points in **a** show the momentum distribution curve at the binding energy indicated by the horizontal dashed line. The double peak structure is attributed to the $\mathcal{C}$ band. (**c**) Energy distribution curves (EDCs) at the S point, normalized at binding energy $\epsilon = E - E_F = -1.8$ eV. (**d,e**) Linear light polarization dependence along the $S - \Gamma_2$ direction at $h\nu = 65$ eV. (**f**) EDCs at the momentum indicated by the vertical dashed lines. In both (**c,f**), the $\mathcal{A}$ and $\mathcal{B}$ bands are indicated by red and grey shading, respectively.

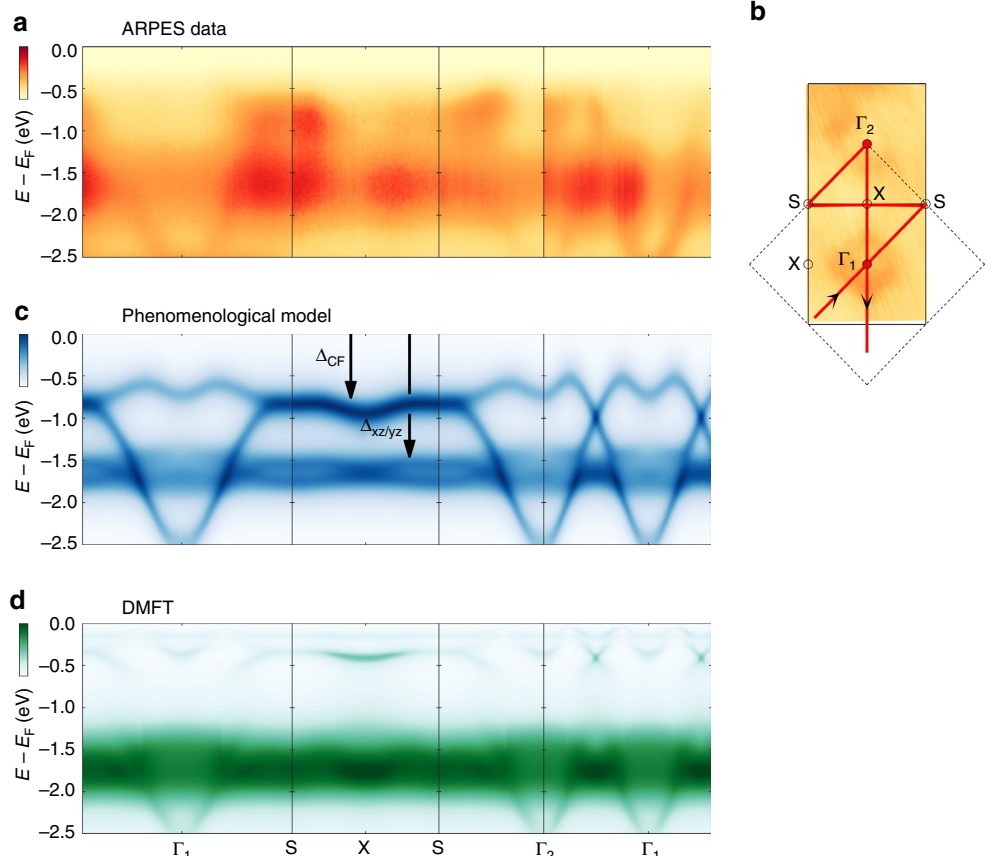

**Figure 3 | Band structure along high-symmetry directions.** (**a**) ARPES spectra recorded along high-symmetry directions with 65 eV circularly polarized light. (**b**) Constant energy map at binding energy $E - E_F = -2.7$ eV. (**c**) DFT-derived spectra for $Ca_2RuO_4$, upon inclusion of a Mott gap $\Delta_{xz/yz} = 1.55$ eV acting on $d_{xz}$, $d_{yz}$ bands and an enhanced crystal field $\Delta_{CF} = 0.6$ eV, shifting spectral weight of the $d_{xy}$ bands (for details, see Methods section) and plotted with spectral weight representation. (**d**) DMFT calculation of the spectral function, with Coulomb interaction $U = 2.3$ eV and a Hund's coupling $J = 0.4$ eV. Dark colours correspond to high intensities.

**Fast dispersing ruthenium bands**. In addition to the flat $\mathcal{A}$ and $\mathcal{B}$ bands, a fast dispersing circular-shaped band is observed (Fig. 3b) around the $\Gamma$-point (zone centre) in the interval $-2.5$ eV $< \epsilon < -2$ eV—see Figs 2a,b and 3a. A weaker replica of this band is furthermore found around $\Gamma_2$ (Fig. 3a,b). The band velocity, estimated from momentum distribution curves (Fig. 2a), yields $v = (2.6 \pm 0.4)$ eV Å. As this band, which we label $\mathcal{C}$, disperses away from the zone centre, it merges with the most intense flat $\mathcal{B}$ band. From the data, it is difficult to conclude with certainty whether the $\mathcal{C}$ band disperses between the $\mathcal{A}$ and $\mathcal{B}$ bands. As this feature is weak in the spectra recorded with 78 eV photons (Fig. 2b), it makes sense to label $\mathcal{A}$ and $\mathcal{C}$ as distinct bands.

**Orbital band character**. Next we discuss the orbital character of the observed bands. As a first step, comparison to the band structure calculations is made. Although details can vary depending on exact methodology, all band structure calculations of $Ca_2RuO_4$ find a single fast dispersing branch[22–25]. Our DFT calculation reveals that the fast dispersing band has predominantly $d_{xy}$ character (Fig. 4a). We thus conclude that the in-plane extended $d_{xy}$ orbital is responsible for the $\mathcal{C}$ band. Within the DFT calculation, the $d_{xz}$ and $d_{yz}$ bare bands are relatively flat throughout the entire zone. This is also the characteristic of the observed $\mathcal{B}$ band. It is thus natural to assign a dominant $d_{xz}$, $d_{yz}$ contribution to this band. The orbital character of the $\mathcal{A}$ band is not obviously derived from comparisons to DFT

calculations. In principle, photoemission matrix element effects carry information about orbital symmetries. As shown in Fig. 2, the $\mathcal{A}$ band displays strong matrix element effects as a function of photon energy and photon polarization. However, probing with 65 eV light, the spectral weight of the $\mathcal{A}$ band is not displaying any regularity within the $(k_x, k_y)$ plane—see Supplementary Fig. 1. The contrast between linear horizontal and vertical light therefore vary strongly with momentum. This fact precludes any simple conclusions based on matrix element effects.

## Discussion

Having explored the orbital character of the electronic states, we discuss the band structure in a more general context. Bare band structure calculations, not including Coulomb interaction, find that states at the Fermi level have $d_{xy}$ and $d_{xz}/d_{yz}$ character (see Fig. 4a). Including a uniform Coulomb interaction $U$ results in a single Mott gap acting equally on all orbitals. Generally, this produces one single flat band inconsistent with the observation of two distinct flat bands ($\mathcal{A}$ and $\mathcal{B}$). Adding in a phenomenological manner orbital-dependent Mott gaps to the self-energy produces two sets of flat bands. For example, one can introduce $\Delta_{xy} = 0.2$ eV and $\Delta_{xz,yz} = 1.5$ eV to mimic the $\mathcal{A}$ and $\mathcal{B}$ bands. However, such Mott gaps are not shifting the bottom of the fast V-shaped dispersion to the observed position. Better agreement with the observed band structure is found, when a Mott gap $\Delta_{xz,yz} = 1.55$ eV is added to the self-energy of the $d_{xz}$, $d_{yz}$ states and a crystal field-induced downward shift $\Delta_{CF} = 0.6$ eV of

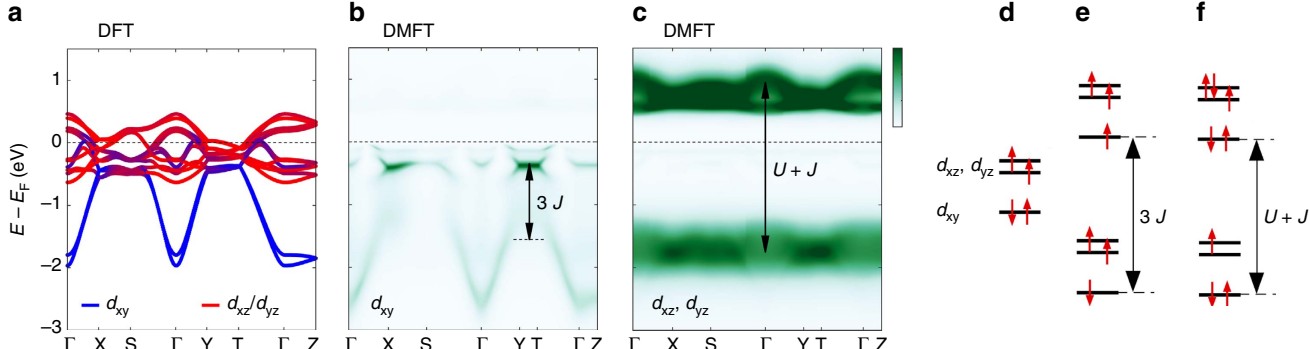

**Figure 4 | Calculated orbital band character.** (**a**) DFT calculation of the bare band structure. $d_{xy}$ and $d_{xz}$, $d_{yz}$ characters are indicated by blue and red colours, respectively. (**b,c**) Are the spectral function calculated within the DMFT approach and projected on the $d_{xy}$ and $d_{xz}$, $d_{yz}$ orbitals, respectively. Dark colours correspond to high intensities. The indicated energy splittings stem from a $t_{2g}$ multiplet analysis in the atomic limit. (**d**) Ground-state multiplet defined by the crystal field and Hund's coupling $J$. (**e**) $d_{xy}$ electron removal configurations split by $3J$ (see main text for explanation). (**f**) Representation of the twofold degenerate $d_{xz}$, $d_{yz}$ electron addition and removal states, split by $U + J$.

the $d_{xy}$ states is introduced. As shown in Fig. 3c, this reproduces two flat bands and simultaneously positions correctly the fast dispersing $\mathcal{C}$ band. From the fact that the bottom of the $\mathcal{C}$ band is observed well below the $\mathcal{B}$ band, we conclude that an—interaction enhanced—crystal field splitting is shifting the $d_{xy}$ band below the Fermi level.

A similar structure emerges from DMFT calculations[26] including $U = 2.3$ eV and Hund's coupling $J = 0.4$ eV. The obtained spectral function (Fig. 3d) is generally in good agreement with the experimental observations (Fig. 3a). Both the $\mathcal{C}$ and $\mathcal{B}$ bands are reproduced with the previously assigned $d_{xy}$ and $d_{xz}$, $d_{yz}$ orbital character (Fig. 4b,c). The $\mathcal{A}$ band is also present in the DMFT calculation around $-0.5$ eV $< \epsilon < 0$ eV. Even though it is not smoothly connected with the $\mathcal{C}$ band, it has in fact $d_{xy}$ character (Fig. 4b). By analysing the multiplet eigenstates (Fig. 4d) and electronic transitions in the atomic limit of an isolated $t_{2g}$ shell, we can provide a simple qualitative picture of both observations: (i) the energy splitting between the $\mathcal{A}$ and $\mathcal{C}$ bands having $d_{xy}$ orbital character, which we find to be of order $3J$, and (ii) the $d_{xz}$ and $d_{yz}$ orbital-driven $\mathcal{B}$ band splitting across the Fermi level, found to be of order $U + J$. Within this framework, the atomic ground state has a fully occupied $d_{xy}$ orbital, while the $d_{xz}$, $d_{yz}$ orbitals are occupied by two electrons with parallel spins ($S = 1$) and thus effectively half-filled. The Mott gap developing in the $d_{xz}$, $d_{yz}$ doublet is thus $U + J$ in the atomic limit[5], corresponding to the electronic transition where one electron is either removed from this doublet or added to this doublet (leading to a double occupancy). In contrast, there are two possible atomic configuration that can be reached when removing one electron out of the fully filled $d_{xy}$ orbital (Fig. 4d). One of these final states (high spin) has $S = 3/2$, $L = 0$ (corresponding pictorially to one electron in each orbital all with parallel spins), while the other (low spin) has $S = 1/2$, $L = 2$ (corresponding to the case when the remaining electron in the $d_{xy}$ orbital has a spin opposite to those in $d_{xz}$, $d_{yz}$). The energy difference between these two configurations is $3J$, thus accounting for the observed ARPES splitting between the two $d_{xy}$ removal peaks. Furthermore, this analysis allows to assess, from the experimental value of this splitting $\sim 1.2$ eV, that the effective Hund's coupling for the $t_{2g}$ shell is of the order of 0.4 eV. This is consistent with previous theoretical work in ruthenates[27,28] and provides a direct quantitative experimental estimate of this parameter. Because the high spin state is energetically favourable with respect to the low spin state (by $\sim 3J$), it can be assigned to the $\mathcal{A}$ band near the Fermi level, while the low spin state can be

assigned to the $\mathcal{C}$ band (See ref. 5 for a detailed description of the atomic multiplets of the $t_{2g}$ Kanamori Hamiltonian). The Hund's coupling has thus profound impact on the electronic structure of the paramagnetic insulating state of $Ca_2RuO_4$. The fact that Hund's coupling mainly influence the $d_{xy}$ electronic states highlights orbital differentiation as a key characteristic of the Mott transition. Moreover, our findings emphasize the importance of the crystal field stabilization of the $d_{xy}$ orbital[19,20]. To further understand the interplay between $U$ and $J$, detailed experiments through the metal–insulator transition of $Ca_{2-x}Sr_xRuO_4$ would be of great interest.

## Methods

**Experimental.** High-quality single crystals of $Ca_2RuO_4$ were grown by the flux-feeding floating-zone technique[29,30]. ARPES experiments were carried out at the SIS, I05 and MAESTRO beamlines at the Swiss Light Source, the Diamond Light Source and the Advanced Light Source. Both horizontal and vertical electron analyser geometry were used. Samples were cleaved *in situ* using the top-post cleaving method. All spectra were recorded in the paramagnetic insulating phase ($T = 150$ K), resulting in an overall energy resolution of approximately 50 meV. To avoid charging effects, care was taken to ensure electronic grounding of the sample. Using silver epoxy (EPO-TEK E4110) cured just below $T = 350$ K (inside the S-Pbca phase—space group 61) for 12 h, no detectable charging was observed when varying the photon flux.

**DFT band structure calculations.** We computed electronic structures using the projector augmented wave method[31,32] as implemented in the VASP[33,34] package within the generalized gradient approximation[35]. Experimental lattice constants ($a = 5.39$ Å, $b = 5.59$ Å and $c = 11.77$ Å) and a $12 \times 10 \times 4$ Monkhorst-Pack $k$-point mesh was used in the computations with a cutoff energy of 400 eV. The spin–orbit coupling effects are included self-consistently. In order to model Mott physics, we constructed a first-principles tight-binding model Hamiltonian, where the Bloch matrix elements were calculated by projecting onto the Wannier orbitals[36,37], which used the VASP2WANNIER90 interface[38]. We used Ru $t_{2g}$ orbitals to construct Wannier functions without using the maximizing localization procedure. The resulting 24-band spin–orbit coupled model with Bloch Hamiltonian matrix $\hat{H}_{\mathbf{k}}^0$ reproduces well the first-principle electronic structure near the Fermi energy. To model the spectral function, we added a gap with a leading divergent $1/\omega$ term to the self-energy $\hat{\Sigma}(\omega) = \hat{P}_{xz,yz}\Delta_{xz,yz}^2/\omega + \mathcal{O}(\omega^0)$. To the Hamiltonian, we added a shift $\hat{H}_{\mathbf{k}} = \hat{H}_{\mathbf{k}}^0 - \hat{P}_{xy}\Delta_{CF}$. $\hat{P}_{xy}$ and $\hat{P}_{xz,yz}$ are projectors on the $d_{xy}$ and $d_{xz}$, $d_{yz}$ orbitals, respectively, while $\Delta_{xz,yz}$ is the weight of the poles and $\Delta_{CF}$ mimics an enhanced crystal field. From the imaginary part of the Green's function $\hat{G}(\mathbf{k}, \omega) = \left[\omega - \hat{H}_{\mathbf{k}} - \hat{\Sigma}(\omega)\right]^{-1}$ with the two adjustable parameters $\Delta_{CF}$ and $\Delta_{xz,yz}$, we obtained the spectral function $A(\mathbf{k}, \omega)$ by taking the trace over all orbital and spin degrees of freedom.

**DFT + DMFT band structure calculations.** We calculate the electronic structure within DFT + DMFT using the full potential implementation[39] and the TRIQS library[40,41]. In the DFT part of the computation, the Wien2k package[42] was used.

The local-density approximation (LDA) is used for the exchange-correlation functional. For projectors on the correlated $t_{2g}$ orbital in DFT + DMFT, Wannier-like $t_{2g}$ orbitals are constructed out of Kohn–Sham bands within the energy window $(-2, 1)$ eV with respect to the Fermi energy. We use the full rotationally invariant Kanamori interaction in order to ensure a correct description of atomic multiplets[5]. To solve the DMFT quantum impurity problem, we used the strong-coupling continuous-time Monte Carlo impurity solver[43] as implemented in the TRIQS library[44]. In the $U$ and $J$ parameters of the Kanamori interaction, we used $U = 2.3$ eV and $J = 0.4$ eV, which successfully explains the correlated phenomena of other ruthenate such as $Sr_2RuO_4$ and $ARuO_3$ ($A = Ca$, $Sr$) within the DFT + DMFT framework[27,28].

**Data availability.** All relevant data are available from the authors.

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

## Acknowledgements

D.S., J.C., C.G.F. and H.M.R. acknowledge support by the Swiss National Science Foundation and its Sinergia network MPBH. Y.S. is supported by the Wenner-Gren foundation. T.-R.C. and H.-T.J. are supported by the Ministry of Science and Technology, National Tsing Hua University, National Cheng Kung University and Academia Sinica, Taiwan. T.-R.C. and H.-T.J. also thank NCHC, CINC-NTU and NCTS, Taiwan for technical support. A.G. and M.K. acknowledge the support of the European Research Council (ERC-319286 QMAC, ERC-617196 CORRELMAT) and the Swiss National Science Foundation (NCCR MARVEL). S.M. acknowledges support by the Swiss National Science Foundation (Grant No. P2ELP2-155357). This work was performed at the SIS, I05 and MAESTRO beamlines at the Swiss Light Source, Diamond Light Source and Advanced Light Source, respectively. We acknowledge Diamond Light Source for time on beamline I05 under proposal SI14617 and SI12926 and thank all the beamline staff for technical support. The Advanced Light Source is supported by the Director, Office of Science, Office of Basic Energy Sciences, of the U.S. Department of Energy under Contract No. DE-AC02-05CH11231. M.K. and A.G. are grateful to M. Ferrero, O. Parcollet and P. Seth for discussions and support.

## Authors contributions

R.F., A.V. and V.G. grew and prepared the $Ca_2RuO_4$ single crystals. D.S., C.G.F., M.S., F.C., Y.S., G.G., M.G., H.M.R., N.C.P., C.E.M., M.S., M.H., T.K.K. and J.C., prepared and carried out the ARPES experiment. D.S., C.G.F., F.C. and J.C. performed the data analysis. T.-R.C., H.-T.J. and T.N. made the DFT band structure calculations. M.K. and A.G. performed and analysed the DMFT calculations. All authors contributed to the manuscript.

## Additional information

**Competing interests:** The authors declare no competing financial interests.

**Publisher's note**: 

