## [Peer Review File · Nature Communications]

Reviewers' comments:

Reviewer #1 (Remarks to the Author):

The physical understanding of transition metal oxides is very often based on simplified models that contain the essential microscopic mechanisms to explain, e.g., the existence of characteristic phase transitions, magnetic order or even superconductivity. In any case the fundamental model parameters, however, require an independent determination either from first principle calculations or from experiment. In the present work, the authors present a detailed study on the paradigmatic Mott insulator Ca_2RuO_4 in the paramagnetic insulating phase, combining the results of angle resolved photoemission spectroscopy experiments together with DFT and DMFT calculations. The central aspect is the explanation of the non-uniform gap, appearing experimentally as two bands (or better: features) A and B with different binding energies below the chemical potential, and therewith the nature of the band gap. In principle, this explanation is rather simple, namely a lifting of the degeneracy of the $\text{Ru}4d$ states.

To estimate the splitting, the authors focus on the importance of the Hund's coupling J , which is of the same order of magnitude as several other material parameters, e.g., the hybridization band width, the crystal electric field (0.6 eV), the Coulomb repulsion energy U (2.3 eV), or finally the observed splitting of the two spectral features itself. The consideration of all these parameters in the DMFT calculation results in a reasonable well description of the experimental results.

Although the authors have structured the paper very carefully and, furthermore, a phenomenological model was introduced to support the assignment of the bands to the respective orbital character, it is very hard to assess the mingling of the many physical parameters. The interpretation given here is not unreasonable and even the stated "first direct quantitative estimate of the effective Hund's coupling from spectroscopy data" could be true. However, it remains unclear whether another parameter set, which possibly would lead to a different interpretation, does not lead to at least a similarly good description. Without ruling out other explanations in the frame of the same physical model, the conclusions remain rather speculative.

Anyway, since the topic is timely and both the data and the calculations are of very high quality, I recommend to publish the present manuscript in Nature Communications.

Reviewer #2 (Remarks to the Author):

The authors report a direct observation of the controversial Mott insulating state realized in Ca_2RuO_4 by ARPES combined with DFT and DMFT calculations. Strong correlation effects on multi-orbital systems are indeed important and hot topics, and Ca_2RuO_4 is one of the most famous but still uncovered materials. Therefore, this study meets the standard of Nat. Commun. in terms of novelty and importance, but, I have one question regarding the conclusion made in this study, which should be resolved.

In page 3, the authors said "The Mott gap, defining the energy scale between lower and upper Hubbard bands, has previously been associated with an activation energy scale ~ 0.4 eV derived from resistivity measurements. Assuming that the Fermi level is centred approximately symmetrically between lower and upper Hubbard bands, our spectroscopic observation is consistent with the transport experiments".

However, the band A is identified to have d_{xy} character and is not regarded as the "lower Hubbard band" in this manuscript. At least in Fig. 4(b), we cannot find any "upper Hubbard band" for d_{xy} (or some small weight exists?). I wonder which interpretations the authors support, "two Mott energy scale (d_{xy} , d_{xz}/yz)" or "band-insulating d_{xy} and half-filled d_{xz}/yz ", because the most parts of this manuscript seem to support the latter but the sentences cited above do not. Note that if one assumes the lowest unoccupied bands to be the upper Hubbard bands of d_{xz}/yz , U might be too small.

I understand that the unoccupied states are out of range for ARPES, but it is directly related to the controversial nature of Mott insulating state of Ca_2RuO_4 , which is a central object in this study. Could the authors make the argument presented in the above-cited sentences clearer?

REVIEWERS' COMMENTS:

Reviewer #1 (Remarks to the Author):

The authors have considered all comments and remarks carefully and improved the manuscript accordingly. I recommend to publish the article as is.

Reviewer #2 (Remarks to the Author):

The authors properly revise the manuscript according to the reviewers' comments. I recommend to publish the present manuscript in Nature Communications.

REPLIES TO THE REFEREES

Referee #1 (Remarks to the Author):

“Anyway, since the topic is timely and both the data and the calculations are of very high quality, I recommend to publish the present manuscript in Nature Communications.”

We thank referee 1 for his/her clear recommendation to publish our manuscript in Nature Communications. Below, we describe how we amended the feedback.

“Although the authors have structured the paper very carefully and, furthermore, a phenomenological model was introduced to support the assignment of the bands to the respective orbital character, it is very hard to assess the mingling of the many physical parameters. The interpretation given here is not unreasonable and even the stated “first direct quantitative estimate of the effective Hund's coupling from spectroscopy data” could be true. However, it remains unclear whether another parameter set, which possibly would lead to a different interpretation, does not lead to at least a similarly good description. Without ruling out other explanations in the frame of the same physical model, the conclusions remain rather speculative.”

The referee gives three suggestions for improvements:

- (1) More clear presentation of physical parameters.**
- (2) Be more careful about the claim of first direct experimental estimate of Hund coupling in the ruthenates.**
- (3) Be more clear on the fact that even a successful theory has been presented it does not rule out all other possible explanations.**

We believe that point (1) applies mainly to the first paragraph of the discussion section. To improve, we have added more description to make it easier to follow. This paragraph now reads:

“Including a uniform Coulomb interaction U results in a single Mott gap acting equally on all orbitals. Generally, this produces one single flat band inconsistent with the observation of two distinct flat bands (A and B). Adding in a phenomenological fashion orbital dependent Mott gaps to the self-energy produces two sets of flat bands. For example, one can introduce $\Delta_{xy}=0.2$ eV and $\Delta_{xz,yz}=1.5$ eV to mimic the A and B bands. However, such Mott gaps are not shifting the bottom of the fast V-shaped dispersion to the observed position.”

Additionally, we have added arrows to figure 3b) to indicate the energy scales according to the main text.

To address (2), we have revised the sentence “... and provides the first experimental estimate ...” so that it now reads: “...and provides a direct quantitative experimental estimate of this parameter.”

For point (3), we have revised the penultimate sentence of the introduction. So that it now reads: "The experimental results, together with our theoretical analysis, elucidate a simple explanation for the Mott phase of the prototypical multi-orbital system Ca_2RuO_4 ." In this fashion, we emphasise that it is "an explanation" rather than "the theory" for this Mott phase.

Referee #2 (Remarks to the Author):

"The authors report a direct observation of the controversial Mott insulating state realized in Ca_2RuO_4 by ARPES combined with DFT and DMFT calculations. Strong correlation effects on multi-orbital systems are indeed important and hot topics, and Ca_2RuO_4 is one of the most famous but still uncovered materials. Therefore, this study meets the standard of Nat. Commun. in terms of novelty and importance, but, I have one question regarding the conclusion made in this study, which should be resolved."

We also thank for his/her recommendation and constructive suggestion.

"In page 3, the authors said "The Mott gap, defining the energy scale between lower and upper Hubbard bands, has previously been associated with an activation energy scale ~ 0.4 eV derived from resistivity measurements. Assuming that the Fermi level is centred approximately symmetrically between lower and upper Hubbard bands, our spectroscopic observation is consistent with the transport experiments".

However, the band A is identified to have d_{xy} character and is not regarded as the "lower Hubbard band" in this manuscript. At least in Fig. 4(b), we cannot find any "upper Hubbard band" for d_{xy} (or some small weight exists?). I wonder which interpretations the authors support, "two Mott energy scale (d_{xy} , d_{xz}/yz)" or "band-insulating d_{xy} and half-filled d_{xz}/yz ", because the most parts of this manuscript seem to support the latter but the sentences cited above do not. Note that if one assumes the lowest unoccupied bands to be the upper Hubbard bands of d_{xz}/yz , U might be too small.

I understand that the unoccupied states are out of range for ARPES, but it is directly related to the controversial nature of Mott insulating state of Ca_2RuO_4 , which is a central object in this study. Could the authors make the argument presented in the above-cited sentences clearer?"

The referee is pointing out an inaccurate statement. Indeed, the activation energy is within our interpretation not related to the lower Hubbard band. We have therefore revised this sentence so that it now reads:

"This energy scale is reasonably in agreement with the activation energy ~ 0.4 eV extracted from resistivity experiments."

In this fashion, we avoid connecting the activation energy to a particular interpretation already in the results section. We thank the referee for his / her careful reading of the manuscript.

Response to the Review reports

REVIEWERS' COMMENTS:

Reviewer #1 (Remarks to the Author):

The authors have considered all comments and remarks carefully and improved the manuscript accordingly. I recommend to publish the article as is.

Reviewer #2 (Remarks to the Author):

The authors properly revise the manuscript according to the reviewers' comments. I recommend to publish the present manuscript in Nature Communications.

We thank both referees for their recommendation to publish this work in Nature Communications.